# Berry Pomace Extracts as a Natural Washing Aid to Mitigate Enterohaemorrhagic *E. coli* in Fresh Produce

**DOI:** 10.3390/foods13172746

**Published:** 2024-08-29

**Authors:** Kanchan Thapa, Dita Julianingsih, Chuan-Wei Tung, Anna Phan, Muhammad Abrar Hashmi, Kayla Bleich, Debabrata Biswas

**Affiliations:** 1Department of Animal and Avian Sciences, University of Maryland, College Park, MD 20742, USA; kthapa@umd.edu (K.T.); djuliani@umd.edu (D.J.); vmtung@umd.edu (C.-W.T.); muhammad.hashmi11@gmail.com (M.A.H.); kbleich@terpmail.umd.edu (K.B.); 2Biological Sciences Program, Molecular and Cellular Biology, University of Maryland, College Park, MD 20742, USA; aphan12@umd.edu

**Keywords:** Enterohaemorrhagic *E. coli*, berry pomace extract, washing solution, cross contamination

## Abstract

Enterohemorrhagic *Escherichia coli* (EHEC) outbreaks have been frequently linked to the consumption of produce. Furthermore, produce grown on organic farms possess a higher risk, as the farmers avoid antibiotics and chemicals. This study sets out to evaluate the effectiveness of advanced postharvest disinfection processes using berry pomace extracts (BPEs) in reducing EHEC load in two common leafy greens, spinach and lettuce. Spinach and lettuce were inoculated with ~5 log CFU/leaf EHEC EDL-933 and then treated with three different concentrations of BPE (1, 1.5, and 2 gallic acid equivalent, GAE mg/mL) for increasing periods of time. After the wash, the bacteria were quantified. Changes in the relative expression of virulence genes and the genes involved in cell division and replication and response against stress/antibiotics were studied. We observed a significant reduction in EHEC EDL933, ranging from 0.5 to 1.6 log CFU/spinach leaf (*p* < 0.05) washed with BPE water. A similar trend of reduction, ranging from 0.3 to 1.3 log CFU/mL, was observed in pre-inoculated lettuce washed with BPE water. We also quantified the remaining bacterial population in the residual treatment solutions and found the survived bacterial cells (~3 log CFU/mL) were low despite repeated washing with the same solution. In addition, we evaluated the phenolic concentration in leftover BPE, which did not change significantly, even after multiple uses. Alterations in gene expression levels were observed, with downregulation ranging from 1 to 3 log folds in the genes responsible for the adhesion and virulence of EHEC EDL933 and significant upregulation of genes responsible for survival against stress. All other genes were upregulated, ranging from 2 to 7 log folds, with a dose-dependent decrease in expression. This finding shows the potential of BPE to be used for sanitation of fresh produce as a natural and sustainable approach.

## 1. Introduction

Foodborne illnesses, a significant threat to public health, have been on the rise in recent years. The most common bacterial foodborne pathogens in the United States (US) include non-typhoidal *Salmonella enterica*, Shiga toxin-producing *Escherichia coli* (STEC), and *Listeria monocytogenes*, which are responsible for an estimated 1.49 million illnesses, 28,000 hospitalizations, and 700 deaths, with an estimated cost of more than USD 6 billion annually according to the Centers for Disease Control and Protection (CDC) in the US [1]. Since 2018, 50 multistate outbreaks of foodborne pathogens have occurred. Of the fifty outbreaks, thirty-eight (76%) had illnesses caused by diarrheagenic *E. coli*, specifically Enterohemorrhagic *E. coli* (EHEC) or non-typhoidal *S. enterica*. More specifically, fruits and vegetables, which are commonly consumed raw, were responsible for at least 15 of these outbreaks [2]. The CDC estimates that each year, EHEC, and particularly *E. coli* O157:H7 alone, results in 63,000 cases of hemorrhagic colitis in the US [1]. Globally, *E. coli* was found to be involved in 2.8 million cases annually in 10 of the 14 subregions recognized by the World Health Organization [3]. In addition to diarrhea, the production of Shiga-like toxin (Stx) by EHEC leads to life-threatening complications, known as hemolytic uremic syndrome (HUS) or renal failure [4,5,6]. The use of conventional antibiotics has been observed to exacerbate Shiga toxin-mediated cytotoxicity, and patients treated with antibiotics for EHEC show a higher risk of developing HUS [7]. This necessitates the development of alternative antimicrobials to address the challenges of conventional antibiotics. In addition, more effective and safer antimicrobials are needed because of the emergence of antibiotic-resistant cases among these pathogens.

Furthermore, researchers have predicted that the growing trend of organic fresh produce consumption might have a direct link to foodborne outbreaks [8,9]. This is due to organic food production or farming practices in which food products are produced without antibiotics or synthetic chemicals, and such practices increase the chance of cross-contamination of the food products with enteric pathogens [10]. In addition, many organic farmers grow their products in integrated crop–livestock systems, where crops are cultivated and livestock is raised on the same farm/facility, and animal gut microbes can be transmitted to plant products through complex farm ecosystems [11].

Green leafy vegetables such as spinach, lettuce, kale, cabbage, and coriander, among many others, are widely consumed because they possess a variety of beneficial micronutrients, which have been shown to lower the risk of several infectious diseases and non-communicable diseases [12]. FAO and WHO suggest the inclusion of several servings of green leafy vegetables in the daily diet for a healthier and balanced diet [13]. However, these foods are available as “ready-to-eat” and are mostly consumed raw, so there is a greater risk of them being vehicles for bacterial transmission. CDC reported that leafy greens, such as lettuce, spinach, and celery, contaminated with EHEC are frequently associated with many foodborne pathogens [1]. In addition, the low infectious dose of EHEC and its ability to survive for prolonged periods in the environment make the contamination of raw and fresh produce with this pathogen increasingly concerning [14,15]. Cattle, the primary reservoir for EHEC, act as a source of cross-contamination with the pathogen into soil and water, which eventually gains access to leafy greens. In addition, the use of untreated/partially composed animal waste or sewage from farms or wastewater effluents for irrigation purposes increases the risk of contamination of fresh produce [16,17]. Moreover, fresh produce is contaminated at different points of production, including harvest, postharvest, storage, processing, and transportation [18,19]. 

To minimize the risk associated with the contamination of leafy greens with *E. coli* O157:H7 and other pathogens, water in combination with chemical disinfectants is commonly used. Although disinfectants and water-based techniques are essential for lowering the microbiological contamination in fresh produce, wash water itself has been shown to be a major source of cross-contamination [20,21]. This is because when fruits and vegetables are washed, microorganisms are released into the water, which then spreads to other produce units, potentially contaminating them [22]. Most chemical washes typically consist of chlorine and peroxyacetic acid compounds. These have been associated with the production of harmful halogenated byproducts, which can directly affect both human health and the environment [23]. Therefore, many current investigations have focused on the search for alternative antimicrobials to be used as wash solutions. As there is an increasing demand for healthy, convenient, and long-lasting ways to improve the safety of fresh produce without sacrificing nutritional value, the use of natural antimicrobial compounds as wash treatments can be the best alternative.

Natural antimicrobials are emerging as an effective and healthier option, with similar properties as chemical antimicrobials. The efficacy of various natural antimicrobials, including berry pomace phenolics, oregano extracts, lactic acid, vinegar, and essential oil nano emulsion, among others, as a postharvest wash for sanitizing fresh produce has been widely studied [10,24,25]. Previously, our lab reported that the byproducts/pomace of blackberry and blueberry contain a wide range of phenolic compounds and organic acids that inhibit several enteric bacterial pathogens [26,27]. Berry pomace, a byproduct of juice industries, is a rich source of phenolic acids, which exhibit antimicrobial activity against a wide range of pathogens [26,28]. Berry pomace extract [BPE] could be a safe, cost-effective, and environmentally friendly method for washing fresh produce to reduce the microbial burden. In this study, we aim to evaluate the potential of berry pomace extracts as a wash solution to reduce the EHEC load on fresh produce. Specifically, our objective is to observe the bacterial reduction in the bacteria-inoculated fresh produce using a BPE-wash treatment of the produce for different time points, as well as its impact on the expression of the vital genes of EHEC, which are important for their survival, replication, and adhesion, in comparison to the wash treatment with control (sterile DI water). 

## 2. Materials and Methods

### 2.1. Bacterial Strains and Growth Conditions

Enterohemorrhagic *E. coli* O157:H7 EDL-933 (EHEC EDL-933, ATCC 700927) was used in this study. The bacteria were grown on Sorbitol MacConkey Agar (Becton, Dickinson and Co., Sparks, MD, USA) at 37 °C, under aerobic conditions (Thermo Fisher Scientific Inc., Waltham, MA, USA), for 24 h. 

### 2.2. Preparation of the Bacterial Inoculum

Packaged spinach (*Spinacia oleracea*) and iceberg lettuce (*Lactuca sativa* var. *capitata*) were purchased from a local supermarket (Giant). Intact healthy lettuce and spinach leaves were cut into 6 cm × 5 cm uniform pieces. The leaves were first washed with deionized [DI] water to remove any microbes or unexpected materials that could be present on the leaves, followed by further disinfection by exposing the leaves under a UV lamp (Labconco, Kansas City, MO, USA) at a wavelength of 254 nm, for 20 min on each side. UV radiation of wavelength ranging from 200 to 280 nm has been shown to possess germicidal properties [29]. Moreover, many studies have reported effective disinfection of fresh produce using UV light [30,31,32], which is why we used a UV lamp to further disinfect the leaves before inoculation.

Overnight grown EHEC colonies were used to prepare a bacterial suspension using phosphate-buffered saline (PBS), and the concentration was measured using a spectrophotometer (Perkin–Elmer, Waltham, MA, USA). The optical density (OD_600_) was fixed at 0.1 [approximately 10^8^ CFU/mL], and the suspension was diluted with PBS to obtain a final bacterial load of 10^5^ CFU/mL (5 log CFU/mL). Each leaf was then immersed in this suspension of EHEC for 20 min for proper inoculation, after which the inoculum from the leaves was determined by serial microdilution using PBS and plate count methods. The inoculum concentration was then determined to be 5 log CFU/leaf after counting the bacterial cells in LB (Luria-Bertani) agar (VWR Chemicals, Solon, OH, USA) after overnight incubation at 37 °C, under aerobic conditions.

### 2.3. Preparation of Berry Pomace Extract and Determination of Phenolic Content

The berry pomace extract was kindly provided by Milne Fruit Products Inc., 804 Bennett Ave Prosser, WA, USA. Briefly, 2.5 g of berry pomace was suspended in 50 mL of 10% ethanol-containing water and incubated overnight at 60 °C. After 24 h, the solid portion of the pomace was separated by centrifugation at 3000× *g* for 20 min, and the solvent was evaporated by leaving it in an oven at 60 °C. Then, the extract was resuspended in deionized water to get the required concentration of wash solution. The concentration of the extract was determined using the Folin–Ciocalteu assay, which measures the total phenolic concentration in the extract in gallic acid equivalent (GAE) mg/mL [33].

Briefly, gallic acid standard solutions were prepared at concentrations of 0, 0.25, 0.5, 1, 1.5, 2, 2.5, and 3 mg/mL. An amount of 20 μL of the standard solutions and an unknown concentration of the berry pomace were taken, into which 1.58 mL water was added. To this, 100 μL of the Folin–Ciocalteu reagent was mixed and incubated for 8 min at room temperature. Following this 20% sodium carbonate (Na_2_CO_3_) was added to each of the solutions and incubated for 30 min at 40 °C in a water bath. Then, spectrophotometric readings at OD at 750 nm wavelength were noted. With the help of these readings, a standard linear curve was obtained as y = 0.4565x + 0.0155, R^2^ = 0.9904. Finally, using this equation from the standard curve, the total phenolic concentration (TPC) of the unknown concentration of BPE was determined. Three different concentrations of the BPE–water mixture were prepared and used as wash solutions—1 GAE mg/mL, 1.5 GAE mg/mL, and 2 GAE mg /mL.

### 2.4. Determination of Minimum Inhibitory/Bactericidal Concentration of BPE against EHEC EDL-933

The broth microdilution method was used to determine the minimum inhibitory concentration (MIC) of the berry pomace extract (BPE), as previously described [34]. The optical density was adjusted to 0.1 at 600 nm, which is approximately 10^8^ CFU/mL. This was then diluted with LB broth and BPE until a final bacterial load of 10^6^ CFU/mL was achieved. Bacterial inoculum (100 μL) of 10^7^ CFU/mL was added to 900 μL of LB broth. To this bacteria–broth suspension, increasing concentrations of BPE, starting at 0 GAE mg/mL and up to 2 GAE mg/mL, were added. After 24 h incubation in LB agar at 37 °C under aerobic conditions, the lowest concentration of BPE with no visible growth when compared with the control (0 GAE mg/mL of BPE) was determined as the MIC. The Minimum Bactericidal Concentration (MBC) was then calculated as the concentration that resulted in 3 logs reduction (99.9%) in growth compared to the control (0 GAE mg/mL of BPE).

### 2.5. Inhibition of EHEC EDL-933 in Fresh Produce Using BPE Supplemented Water

Pre-inoculated spinach and lettuce were disinfected by soaking and manually swirling them in the respective wash solution [BPE-supplemented water solution and water alone]. Fresh sterile water (control) and three different concentrations of BPE (1, 1.5, and 2 GAE mg/mL) were used to treat the pre-inoculated produce for 15, 30, 45, and 60 s. After each wash, the leaves were suspended in PBS and vortexed to dislodge the bacteria. Aliquots from this PBS solution were used to perform a microdilution plate assay. Plate counts were performed on LB agar, with colonies counted after overnight incubation at 37 °C to determine if bacterial reduction had occurred.

For the first part of this study, we used a single batch of BPE solutions, each with varying concentrations, to wash the produce at four time points. Similarly, aliquots of the residual wash after four soaks (as a result of washing at four time points) were taken, and the bacterial population was quantified to determine the microbial load harbored in the residual solution. However, for the second part of this study, different batches of new BPE-water were used for each time point, after which the leftover wash solution was used to quantify residual bacteria.

After each wash interval, the phenolic acid concentration of the residual solution was measured as GAE mg/mL following the Folin–Ciocalteu procedure. Bacterial counts after BPE treatment for different time points were compared with those of the control and expressed by determining the significant difference between the two. All the experiments were performed in three technical and biological replicates.

### 2.6. RNA Extraction and cDNA Synthesis

RNA extraction was performed following the method described previously [10]. Briefly, spinach and lettuce pre-inoculated with EHEC EDL-933 were treated with different concentrations of BPE-containing water or only water (control) for 60 s. One mL of each of these washing waters was collected in a 2 mL tube and centrifuged at 14,000× *g* for 15 min. Discarding the liquid, the harvested bacterial pellets were lysed with 1 mL of TRIzol^®^ reagent (Molecular Research Center Inc., Montgomery Rd, Cincinnati, USA) for 5 min, at room temperature. Then 200 μL of chloroform (VWR International, Radnor, PA, USA) was added to the lysates and vortexed, followed by incubation at room temperature for 3 min. The samples were then centrifuged at 14,000× *g* for 15 min. To the obtained aqueous solutions, 500 μL of isopropanol [Fisher Scientific Co., Fair Lawn, NJ, USA] was added and incubated at room temperature for 10 min. Centrifugation was repeated at 14,000× *g* for 15 min. Then, after discarding the supernatant, the pellet containing the RNA was re-suspended and vortexed in 1 mL of 75% ethanol for 10 s, then centrifuged at 7500× *g* for 5 min. The supernatant was discarded, and the pellet was air-dried to remove the remaining ethanol and finally dissolved in 50 μL of RNase-free water. The RNA concentration was measured using a Nanodrop spectrophotometer (Thermo Fisher Scientific Inc., Marietta, OH, USA). Then, cDNA was synthesized according to the manufacturer’s instructions, using the High-Capacity cDNA Reverse Transcription Kit [Applied Biosystems, Foster City, CA, USA]. An amount of 10 μL of RNA was mixed with 2 μL of 10xRT Buffer, 0.8 μL of 25x dNTP Mix (100 mM), 2 μL of 10xRT random primers, 4.2 μL of nuclease-free water, and 1 μL of reverse transcriptase. The thermocycler was set to incubate at 25 °C for 10 min, at 37 °C for 120 min, and at 85 °C for 5 min to obtain the cDNA. cDNA was standardized by first determining the concentration of DNA using a nanodrop spectrophotometer and then diluted with molecular water to obtain 100 ng.

### 2.7. Determination of the Expression of Genes Due to BPE Treatment Using qRT-PCR

The qRT-PCR reaction mixture was prepared using 2 μL of the cDNA (100 ng), 2 μL of each primer (100 mM) of the target genes of interest, 6 μL of RNase-free water, and 10 μL of the PerfeCTa SYBR Green Fast Mix [Quanta Bio, Beverly, MA, USA]. The reaction was performed using the Eco Real-Time PCR system (Illumina, San Diego, CA, USA) with 30 s denaturation at 95 °C, followed by 40 cycles of 95 °C for 5 s, 55 °C for 15 s, and 72 °C for 10 s. The relative gene expression was determined by comparative log fold changes. The cycle threshold [Ct] values of the respective genes in bacterial cells from the BPE-treated and water-treated samples were compared to those in untreated bacteria and normalized to the housekeeping gene (16S RNA). The quantitative RT-PCR assay was conducted in triplicate. The primers sequences for the EHEC EDL-933 genes (Erofins MWG Operon, Huntsville, AL, USA) are listed in Table 1.

### 2.8. Statistical Analyses

Significant differences (*p* < 0.05) for each treatment, compared with the control, were statistically analyzed using Microsoft Excel. The Student’s *t*-test was used to determine the significant differences between treatment and control (water), as well as between berry pomace extract-supplemented water and control. Analysis using the *t*-test was performed to determine significant difference in gene expression between treatment with water (control) and each of the BPE treatment groups. The error bars in the figures were calculated as the standard deviation from three replicates.

## 3. Results

### 3.1. MIC and MBC of BPE against EHEC EDL-933

The antimicrobial activity of berry pomace phenolic extracts, BPE, against EHEC EDL-933 was evaluated in the LB broth culture by the plate count method. The MIC of BPE against EHEC was 1 mg GAE/mL. No visible growth was observed in the presence of 1 GAE mg/mL BPE or higher concentrations after overnight incubation at 37 °C aerobically. In comparison to the bacterial count in the control, i.e., 8.4 log CFU/mL, the bacterial count of 5.5 log CFU/mL was obtained at 1 GAE mg/mL after 24 h of incubation. As this concentration of BPE (1 mg GAE/mL) could inhibit the growth of EDL-933 by 3 logs (99.9%), it was considered both the MBC and MIC value. It was also observed that the inhibition of EHEC EDL-933 growth was affected by BPE in a concentration-dependent manner. The ratio of MBC to MIC was calculated to be 1. Since a ratio <2 suggests that the drug is bactericidal, >16 suggests that the bacteria are bacteriostatic, while >32 suggests that the bacteria are tolerant to the drug, we can conclude that BPE possesses bactericidal properties [42].

### 3.2. Inhibition of EHEC by Using a Single Batch of BPE-Containing Water at Various Time Points

The reduction in the EHEC EDL-933 load in pre-inoculated spinach and lettuce was compared between washing with only water (control) and BPE water. With an inoculum of ~10^5^ CFU/leaf, spinach washed with water alone revealed bacterial reductions of 0.2, 0.6, 0.85, and 0.83 log CFU/leaf at 15, 30, 45, and 60 s of treatment, respectively. However, washing with 1 GAE mg/mL BPE-containing water resulted in a decrease by 0.5, 0.6, 0.9, and 1.1 log CFU/leaf at the same time intervals. Increasing the concentration of BPE [1.5 mg GAE/mL] in water increased the reduction rates (*p* < 0.05) by 0.7, 0.71, 1.2, and 1.4 log CFU/leaf after 15, 30, 45, and 60 s of treatment. Similarly, pre-inoculated spinach washed with 2 GAE mg/mL BPE-containing water showed a remarkable reduction in EHEC EDL-933 [*p* < 0.05] by 0.9, 1. 07, 1.8, and 1.6 log CFU per leaf at 15, 30, 45 and 60 s of treatment, respectively (Figure 1A).

Likewise, lettuce harboring a bacterial load of approximately 5 log CFU/leaf, when washed with only water for 15, 30, 45, and 60 s, resulted in a reduction by 0.09, 0.3, 0.5, and 0.56 log CFU/leaf, respectively. In contrast, 1 GAE mg/mL BPE wash led to a decrease by 0.3, 0.37, 0.57, and 0.56 log CFU/leaf at the same time points. The washing of lettuce with 1.5 GAE mg/mL BPE caused a 0.56, 0.64, 1.1, and 1.04 log reduction within the same intervals. Furthermore, with increasing the BPE concentration to 2 mg GAE /mL, a more pronounced decrease (*p* < 0.05) of 0.5, 0.72, 1.15, and 1.3 log was observed at 15, 30, 45, and 60 s, respectively (Figure 1B). These observations indicate a concentration and time-dependent reduction in the bacterial population following the BPE-containing water treatment. Interestingly we observed that the reduction in bacterial loads was greater in spinach, compared to lettuce leaves. However, the overall bacterial inhibition compared to water was not highly significant, which can be attributed to shorter treatment time periods.

In addition, the bacterial content of the remaining wash solutions after four soaks of both spinach (Figure 2A) and lettuce (Figure 2B) was determined. The microbial load in the remaining water (control) was approximately 3 log CFU/mL, indicating 100% viable bacteria (Figure 2). The viable bacterial concentration observed in 1, 1.5, and 2 GAE mg/mL residual BPE water was expressed as the relative percentage of surviving bacteria compared to that of control. In spinach, aliquots of residual BPE water from treatments with 1, 1.5, and 2 GAE mg/mL showed a bacterial load of 71.5%, 52.69%, and 25.89%, respectively, relative to control. On the other hand, in lettuce washed with residual water, the viable bacteria counts were 44.08%, 32.95%, and 20.70% in 1, 1.5, and 2 GAE mg/mL BPE water solutions, respectively. The bacterial load in these treatment solutions was significantly lower compared to the control (*p* < 0.05).

### 3.3. Inhibition of EHEC by Using a New Batch of BPE at Each Time Point

A new batch of BPE water was used for washing the pre-inoculated lettuce and spinach at each time point to compare the maximum effectiveness of BPE water with the appropriate time of dipping. By using single BPE wash solutions for multiple soaks for one trial and a new BPE wash solution for each time point, we aimed to look at approximately how much bacteria would be present in the leftover BPE disinfectant solution and find out whether changing BPE after every batch of produce wash would be more effective in reducing the bacterial load. This also helps determine if BPE can be responsible for cross contamination.

In this experiment, the initial inoculation level of EDL-933 was approximately 5 log CFU/leaf for both the spinach and lettuce samples. The reduction in EDL-933 with water wash observed in the pre-inoculated spinach was 0.27, 0.71, 0.89, and 0.27 logs at times 15, 30, 45, and 60 s, respectively. However, washing water supplemented with 1 GAE mg/mL BPE-containing water yielded a higher reduction of 0.65, 1.23, 1.34, and 1.92 logs at the corresponding time points. At these similar time points, treatment with 1.5 GAE mg/mL BPE-supplemented water resulted in a substantial decrease in the EDL-933 load in spinach by 0.84, 1.24, 1.79 and 1.97 log CFU/leaf. Finally, 2 GAE mg/mL BPE water inhibited the EDL-933 load in spinach maximally by 0.92, 1.84, 2.00, and 2.11 log CFU/leaf at 15, 30, 45, and 60 s, respectively (Figure 3A).

The pre-inoculated lettuce washed with water alone led to bacterial decrease of 0.42, 0.98, 1.24, and 1.31 logs for the time points 15, 30, 45, and 60 s, respectively. On the other hand, treatment with 1 mg GAE /mL BPE showed a reduction in bacterial concentration of 0.45, 1.20, 1.60, and 1.51 logs CFU/leaf at times 15, 30, 45, and 60 s, respectively. Similarly, increasing the BPE concentration in water to 1.5 GAE mg/mL enhanced the reduction in bacterial load by 0.65, 1.26, 1.53, and 1.83 logs, while 2 GAE mg/mL BPE caused a remarkable decrease of 0.73, 1.49, 1.68, and 1.76 log CFU/leaf at time points 15, 30, 45, and 60 s, respectively (Figure 3B). Compared to the reduction achieved using the similar BPE-containing water for varying wash intervals, the reduction with separate BPE solutions was notably higher. This implies that BPE water accumulates bacteria over time, and after multiple uses, it should be replaced to avoid cross-contamination.

### 3.4. Remaining Phenolic Content of BPE after Subsequent Wash

The total phenolic content of the remaining BPE in water after each wash interval was measured using the Folin–Ciocalteu assay. The original concentration of the BPE before the wash was noted and compared with the concentration of remaining BPE in water obtained following the wash at each time point. When washing spinach leaves, the original concentration of 1 GAE mg/mL BPE was reduced to 0.90, 0.71, 0.69, and 0.69 after 15, 30, 45, and 60 s, respectively. In contrast, the concentration decreased from 1.50 GAE mg/mL to 1.42, 1.46, 1.44, and 1.25 GAE mg/mL at the same time points. Finally, the concentration reduced from 2 GAE mg/mL to 1.85, 1.77, 1.75, and 1.57 GAE mg/mL at 15, 30, 45, and 60 s, respectively (Figure 4A).

On the other hand, the lettuce wash resulted in a reduction in the phenolic concentration of BPE in water from 1 GAE mg/mL to 0.83, 0.83, 0.97, and 0.72 mg GAE/mL, respectively, at time points—15, 30, 45, and 60 s. Likewise, the concentration of phenolic compounds in BPE-containing water decreased from 1.5 GAE mg/mL to 1.46, 1.40, 1.33, and 1.25 GAE mg/mL at the same time points, while the concentration of phenolic compounds in 2.16 mg GAE /mL BPE water reduced to 2.14, 1.81, 1.79, and 1.74 GAE mg/mL at 15, 30, 45, and 60 s, respectively (Figure 4B). The reduction observed after each wash was not significant compared to the original concentration, which suggests that the number of washes does not affect much the reduction in the phenolic concentration of the BPE in water.

### 3.5. Alteration of EHEC EDL-933 Gene Expressions in Response to BPE Treatment

Changes in the relative expression of genes related to attachment, adhesion, biofilm formation, survival, and replication of EHEC EDL-933 were assessed following the 60 s wash in BPE. A quantitative real-time PCR was utilized to evaluate the levels of specific genes that were normalized using the 16S rRNA gene as a reference. The fold change values were calculated by comparing gene expression in bacteria treated with varying concentrations of BPE to that of the control group treated with water in both spinach (Figure 5) and lettuce (Figure 6). The relative expression of the genes responsible for attachment to surfaces and adhesion (*espA*, *csgA*, *csgD*, *ecpA*), biofilm formation and motility (*fliC*, *fliA*, *flhDC)*, cell division and replication (*murA*, *holE*, *ftsZ*), and response against stress/antibiotics (*rpoS*, *sodB*, *marA*) due to BPE treatment were studied.

We observed that two genes, *espA* and *holE*, (*p* < 0.05) of EHEC EDL-933 were downregulated significantly due to the BPE treatment [60 s] of pre-inoculated spinach, while the expression of three genes *ftsZ*, *fliC*, and *ecpA* was downregulated, but the downregulation was not statistically significant (*p* > 0.05). On the other hand, the rest of the genes were upregulated. More specifically, the expression of four genes (*rpoS*, *fliA*, *csgA*, and *csgD*) was significant (*p* < 0.05) due to the BPE treatment (60 s) of pre-inoculated spinach. Similarly, in the treated lettuce, *espA*, *holE*, and *fliC* were significantly downregulated (*p* < 0.05), and all the other genes were upregulated. We observed a notable variation in gene expression levels from bacteria harvested from the lettuce and spinach wash solutions.

## 4. Discussion

Vegetables, specifically leafy greens, are one of the most common reservoirs for Enterohemorrhagic *E. coli*, being commonly contaminated through water or soil at the pre-harvest level, at various points of production [43,44]. Chlorine-based sanitizers are widely utilized in fresh produce production chains to remove the pathogenic microbes; however, concerns about chlorine or bleach residues on the product or their effectiveness at acceptable concentrations have prompted an alternative search [45]. Some alternatives to chemical disinfectants include ozonated water and electrolyzed water, which have been found to effectively disinfect fresh fruits and vegetables, with a remarkable reduction in spoilage micro-organisms [46,47]. However, there are some limitations of these techniques, such as variations in control, potential damage to the palatability of the product, and the possibility of developing resistance against the pathogens. Likewise, hydrogen peroxide is a widely used sanitizing chemical agent which is only toxic to pathogens and possesses potent bacteriostatic and bactericidal action, but it requires higher concentration for efficacy, which has been hypothesized to affect product quality [48]. Plant-derived antimicrobials are also emerging as healthier, cheaper, and environmentally friendly substitutes, with berry pomaces as sources of phenolics, flavanones, and anthocyanins being one of them. The exact antimicrobial mechanism of the compounds in berries has not been identified yet, but it has been found that these compounds can inhibit the growth of some Gram-negative bacteria. This has been documented to be due to the permeability of these compounds through the outer membrane of Gram-negative bacteria, which otherwise would confer resistance against conventional antibiotics; hence, these compounds can enter the bacterial cytoplasm, leading to acidification and death [28,49,50,51]. Numerous studies demonstrate the potent inhibitory action of blueberry and blackberry pomaces against several pathogenic bacteria such as *Staphylococcus*, *Listeria*, *Salmonella*, Enterohemorrhagic *E. coli*, and *Campylobacter* [10,26,51,52,53]. In agreement with previous studies, this study showed a significant inhibition of EHEC EDL-933 in fresh produce after washing with BPE-containing water. Furthermore, the phenolic contents of BPE also altered the expression of various genes of EHEC EDL933, which are involved in survival, multiplication, adhesion, and production of toxin, during the dipping treatment at various time points.

The observed inhibitory effect of BPE on EHEC EDL933 growth was confirmed through the determination of the minimal inhibitory concentration (MIC) and minimal bactericidal concentration (MBC) values. We observed both the dose and the duration of treatment time of BPE vastly influenced the reduction of this pathogenic bacterial load in leafy greens. Our findings indicate that even a dose as low as 1 GAE mg/mL of BPE can mitigate bacterial contamination within a minute of treatment. In contrast to other antimicrobials, particularly chlorine or chemicals, the residual effect of BPE is not a concern, as the taste or smell imparted by berries are palatable and its constituents, mostly antioxidants, flavonoids, and polyphenolics, are safe for human consumption.

In this study, the bacterial inoculation of fresh produce was performed by the immersion of leaves in bacterial suspension for 20 min. However, this duration may not have been sufficient for the bacteria to firmly adhere to the plant cells, as they would under natural conditions. It has been observed that EHEC cell attachment would be better investigated in long-term-exposure studies compared to short-term ones [54]. This can be attributed to the easier dislodging of bacteria during the wash treatment. Numerous studies have demonstrated that leaves bearing a smooth surface harness fewer bacteria, while leaves with irregular folds or cut surfaces provide a suitable environment for the bacteria to internalize within the plant cells and protect itself against inactivation by antimicrobial compounds and enable bacterial survival [54,55,56,57]. In this study, fresh and intact spinach leaves and lettuce were used, which could have facilitated the easier dislodging of the bacteria from leaves into the wash solution. This has been corroborated in our study, where the bacterial load reduction was higher in spinach as compared to lettuce. Likewise, the population of viable bacteria in the residual solutions was observed to be greater in the residual wash of spinach in comparison to lettuce. This suggests that the physiological features of lettuce maintained denser microbial populations compared to spinach, so the bacterial adhesion to lettuce cells might have been relatively stronger, as supported by our findings. The bacterial load in the residual water and BPE, which were newly prepared for each time point, was substantially lower compared to when the same wash solutions were reused for repeated time points. This suggests that BPE possesses antimicrobial properties, even after multiple uses; however, it needs to be sterilized before it can be used again. Sterilization can be performed by pasteurization or by filtering the used solution using a 0.2 μm filter. Further confirmation can be obtained by culturing the newly sterilized pomace in an agar medium. However, a point to be noted is that the sterilization technique should not alter the phenolic concentration of the BPE.

Interestingly, after several washes, the bacterial concentration was still remarkably low, although the initial inoculum was 5 log CFU. This implies that water containing BPE has a significant antibacterial activity and is unlikely to cross-contaminate fresh product batches. Further, bacteria that accumulate in the wash treatments have been shown to remain viable without replicating for extended periods of time, which accounts for the existence of live bacteria in the wash solutions that survive after treatment [10,23]. The phenolic concentration of the BPE did not change significantly even after several wash periods, which indicates the reusability of the berry pomace extract solutions. Similar observations were made in a study by Alvarado-Martinez et.al., 2020, where they studied *Salmonella* inhibition, ranging from 0.2 to 5 logs CFU, in fresh produce, using berry pomace polyphenolics and observed no significant changes in berry pomace phenolic concentration after multiple uses [10].

The gene expression analysis revealed significant changes in vital genes associated with EHEC EDL933 pathogenicity and survival in response to BPE treatment. We assessed the genes responsible for adhesion to host cells, such as *espA*, *E. coli* common pilus (*ecpA*), and the curli fiber subunits *csgA* and *csgD*. *espA* is a filamentous apparatus in the Type III Secretion System (TTSS) that have been shown to play a significant role in the colonization of human and bovine hosts. It was found to use the same mechanism to attach to the leaf epidermis of salad leaves such as arugula, spinach, and lettuce [36]. *espA* has been significantly downregulated in both spinach and lettuce after pomace water treatment, which indicates the potential of BPE to reduce EHEC EDL933 attachment to plant surfaces. The role of curli subunits (*csgA* and *csgD*) for biofilm formation and attachment to leafy greens has been well documented [58,59,60]. *ecpA* is another fimbrial structure on EHEC that is also responsible for biofilm formation and adhesion to contact surfaces. Several studies have reported the ability of plant-derived compounds to inhibit the biofilm-related genes [37,60]. Essential oil compounds such as clove oil, pimento oil, eugenol, and a phytochemical compound, lawsone, have been shown to inhibit the expression of the curli genes (*csgABDGF*), along with the inhibition of some motility-associated genes, such as *fliA*, *flih*, and *espD* [37,60]. A downregulation of *ecpA* gene following BPE treatment was noted; however, interestingly, *csgA* and *csgD* were upregulated. Studies have shown alterations of curli expression under unfavorable environments, such as changes in osmolarity, nutrient deprivation, and low temperatures, which could explain why we saw fluctuations in these genes in our study [61,62]. One such study showed that high levels of intracellular *N*-acetylglucosamine-6-phosphate induced defects in curli production [61]. Bacterial cells, including EHEC, secrete a variety of extracellular structures, such as curli fibers, flagella, pili, among others, to adapt to their environments; therefore, differences in the transcription levels of these genes could be linked to exposure to sanitizers such as BPE. Flagellar gene expression was studied via the *flhDC*, *fliC*, and *fliA* genes, which are majorly responsible for motility and biofilm formation in some cases [41]. The downregulation of *fliC* in our study suggests the anti-virulence property of the pomace extracts. *fliA* and *flhDC*, on the other hand, were upregulated, with a decreasing trend as the BPE concentrations increased. We hypothesized that the upregulation of these genes represents a survival response of the bacteria to short-term stress.

Another important gene, *ftsZ*, encodes a tubulin homologue protein that forms the septum, which plays a pivotal role in cell division [34]. *ftsZ* was found to be suppressed in aliquots from spinach and lettuce, which could be due to the action of phenolic bioactive compounds in the pomace extracts, which are suggested to affect the cell membranes and microbial enzymes [63,64]. Surprisingly, *murA*, which is responsible for bacterial cell wall peptidoglycan synthesis, is upregulated. This is possibly due to the survival response of bacteria against stress. In contrast, *holE*, a gene encoding DNA the polymerase III subunit θ [theta], which is vital for bacterial replication, was significantly downregulated. The gene *marA’s* function is to regulate the efflux and influx of toxic compounds/antibiotics by regulating porin synthesis [65]. The upregulation of this gene could indicate the bacterial cells’ response to efflux the berry pomace extract [65,66]. The other two genes, *rpoS*, encoding an alternative sigma factor critical for stress response, and *sodB*, encoding iron superoxide dismutase, displayed upregulation. Previous studies have reported that the application of sublethal stress to bacterial cells produced a signaling response to help them survive. This mechanism could be a possible reason for the upregulation of both *rpoS* and *sodB* genes in treated spinach and lettuce [11]. There were notable differences in observations between spinach and lettuce, which could be due to differences in the physiology of plant cells, the internalization of the pathogens by the plants, the potential defense of plant cells against bacterial pathogens, and other factors, which remain unclear in our study. This warrants further exploration in that direction.

## 5. Conclusions

The antimicrobial activity of BPE against EHEC EDL933 suggests the potential of consumable plant phenolics from berries in producing safer produce, specifically spinach and lettuce, through effective postharvest washing. In addition, the presence of BPE in the washing water alters essential genes of EHEC EDL933, providing insights into how plant phenolic inactivate the growth and survival of this bacterial pathogen. Future studies can include better understanding of bacterial attachment and the internalization of pathogens in various plant produce to develop and optimize different wash techniques or wash periods for bulk samples. Although berry pomace extract is a healthier and more sustainable disinfecting agent, it is less effective compared to chemical-based disinfectants. This limitation can be overcome by increasing the BPE concentration in the wash solution, as well as prolonging the treatment time periods. As a byproduct of juice-processing industries, it is cheaper, although high-scale production could be costly, which can be a potential limitation of using only BPE as a washing aid for fresh produce.

## Figures and Tables

**Figure 1 foods-13-02746-f001:**
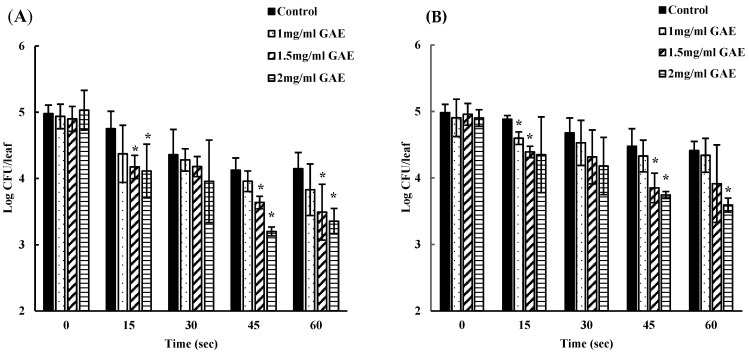
Time-dependent bacterial inhibition by different concentrations of BPE on spinach (**A**) and lettuce (**B**) compared to water (control). Count at 0 s denotes initial inoculum. ***** denotes a statistically significant (*p* < 0.05) difference between treatment with water (control) and BPE.

**Figure 2 foods-13-02746-f002:**
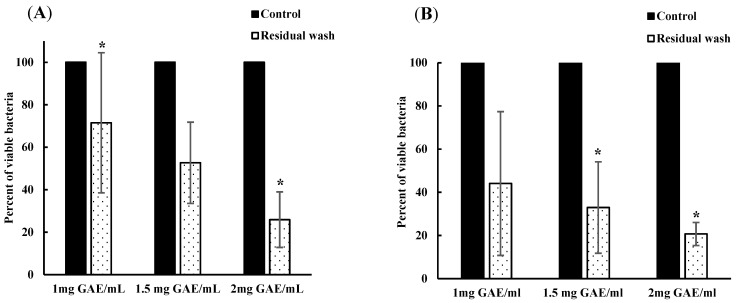
The percentage of viable bacteria in residual BPE-supplemented water after four soaks of spinach (**A**) and lettuce (**B**) compared to that in control [water]. ***** denotes a statistically significant (*p* < 0.05) difference between treatment with water (control) and BPE.

**Figure 3 foods-13-02746-f003:**
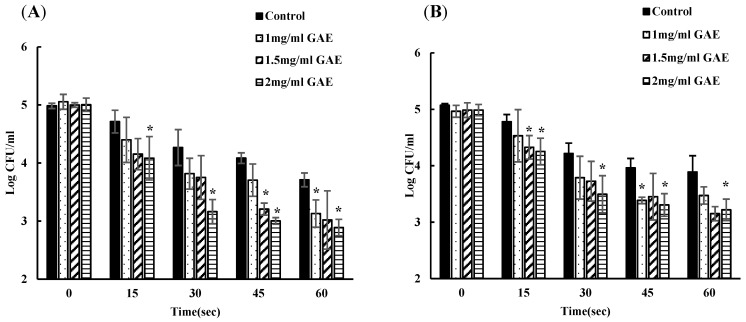
Time-dependent bacterial inhibition by different concentrations of BPE on spinach (**A**) and lettuce (**B**) compared to water (control). Separate fresh BPE was used for different time points. Count at 0 s denotes initial inoculum; ***** denotes a statistically significant (*p* < 0.05) difference between treatment with water (Control) and BPE.

**Figure 4 foods-13-02746-f004:**
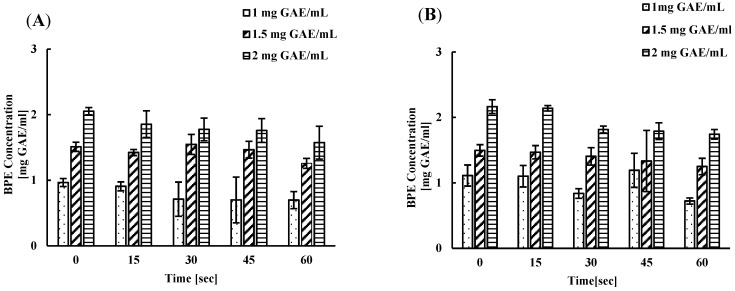
Measurement of concentration of total phenolic compounds (GAE mg/mL) in BPE after subsequent washing spinach (**A**) and lettuce (**B**) leaves at different time points.

**Figure 5 foods-13-02746-f005:**
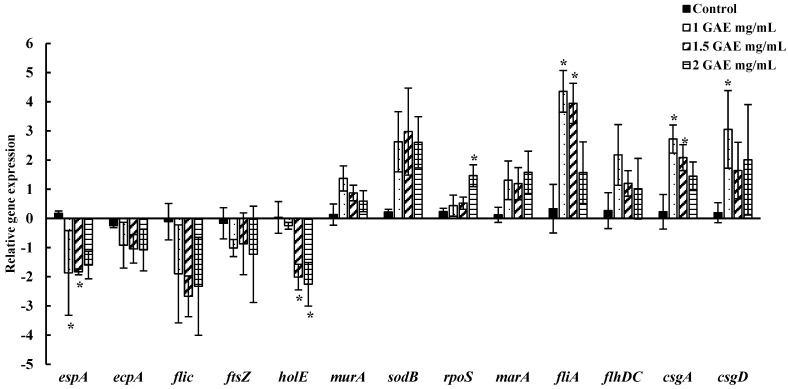
Relative expression of genes associated with adhesion, survival, and virulence proteins in EHEC from spinach treatment for 60 s. ***** indicates a statistically significant (*p* < 0.05) expression change in treatment groups compared to water (control), after normalization to the housekeeping gene (16S rRNA).

**Figure 6 foods-13-02746-f006:**
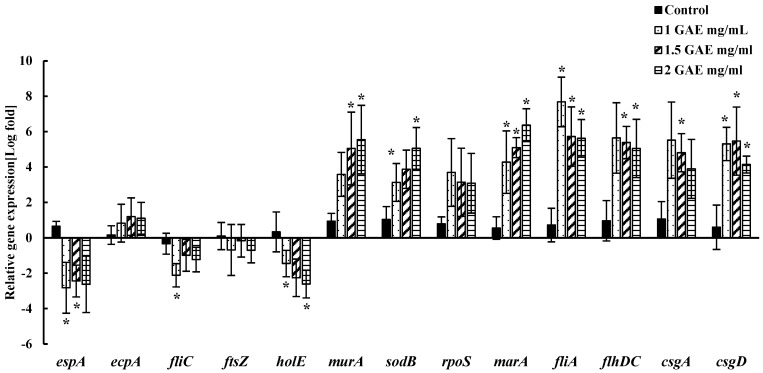
Relative expression of genes associated with adhesion, survival, and virulence proteins in EHEC from lettuce treatment for 60 s. ***** indicates a statistically significant (*p* < 0.05) expression change in treatment groups compared to [water] control, after normalization to the housekeeping gene (16S rRNA).

**Table 1 foods-13-02746-t001:** Primers used to determine gene expression of Enterohemorrhagic *E. coli*.

Gene	Primer Sequence [5′-3]	Function	Reference
16s RNA	F: CGTTACCCGCAGAAGAAGC R: GTGGACTACCAGGGTATCTAATCC	Housekeeping gene	[35]
*espA*	F: GTGAGCAGAGAGAGAATGC R: GTAAATCCAGCGATAAACCCG	Adherence and colonization	[36]
*ecpA*	F: CGCGGATCCATGAAAAAAAAGGTTCTGGC R: CGCGAATTCTAACTGGTCCAGGTCGCGTCG	*E. coli* common pilus	[37]
*fliC*	F: TACCATCGCAAAAGCAACTCCR: GTCGGCAACGTTAGTGATACC	Flagellar filament protein	[35]
ftsZ	F: CTTCTCTTGACCCGGATATG R: CATTCACGACTTTAGCAACC	Cell division	[34]
*holE*	F: ACGTGAACAGCCTGAACATTTGCG R: TGGGCTCGTAAGGTAAACGTGACA	DNA polymerase III subunit θ	[38]
*murA*	F: AACGAAGCTCCAGGGCGAAG R: TTCGCACCCAGCTGGCTTAG	Peptidoglycan synthesis	[39]
*sodB*	F: GCGATCAAAAACTTTGGTT R: CCAGAAGTGCTCAAGAT	Multi-defense system against oxidative stress	[38]
*rpoS*	F: CGCCGGATGATCGAGAGTAA R: GAGGCCAATTTCACGACCTA	Survival against stress	[38]
*marA*	F: TTAGGCCAATACATCCGCAG R: AAGGTTCGGGTCAGAGTTTG	Antibiotic efflux system	[38]
*fliA*	F: TTAGGGATCGATATTGCCGATT R: CGTAGGAGAAGAGCTGGCTGTT	Flagellar biosynthesis sigma factor	[40]
*flhDC*	F: ACAACATTAGCGGCACTGAC R: AGAGTAATCGTCTGGTGGCTG	Master regulator for flagellar synthesis	[41]
*csgA*	F: AGATGTTGGTCAGGGCTCAG R: CGTTGTTACCAAAGCCAACC	Curli subunit	[38]
*csgD*	F: CCGCTTGTGTCCGGTTTT R: GAGATCGCTCGTTCGTTGTTC	Curli subunit	[37]

## Data Availability

The original contributions presented in the study are included in the article, further inquiries can be directed to the corresponding author.

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
