# Peer review of "Berry Pomace Extracts as a Natural Washing Aid to Mitigate Enterohaemorrhagic E. coli in Fresh Produce"

_foods, 2024, doi:10.3390/foods13172746_

Round 1

Reviewer 1 Report

Comments and Suggestions for Authors

The article entitled “Berry pomace extracts as a natural washing aid to mitigate enterohaemorrhagic E. coli in fresh produce.” by  Thapa K.et al., evaluate the adeptness of advanced postharvest disinfection processes using berry pomace extracts to reduce Enterohemorrhagic Escherichia coli load in two common leafy greens, spinach, and lettuce. The article is interesting, well worked and contains complex information regarding the intended purpose.  However, there some changes required, as reported below:

- a graphical abstract would be very helpful for better understanding of the article. Please insert one.

-The introduction part can be improved: -it would be good to add a phrase about general properties and benefits of spinach and lettuce

   - why did you choose the two greens? As we well know, there are other "ready-to-eat" greens that are less studied

The part of material and methods: - what kind of sterile water was used to wash the leaves? Please mention the data about this water!

- Please mention the data about the phate-buffered saline (PBS).

- Please describe  how you got it the blueberries (Vaccinium corymbosum) pomace/byproduct

- It would be very helpful to have a conclusion section where you can see exactly what the goal was and the results reached, maximum 3-4 sentences

Author Response

Please see the attachment below. Response for both the reviewers's comments is present in the single file. Thank you.

Reviewer 2 Report

Comments and Suggestions for Authors

Manuscript 3138349

Journal Foods

Title Berry pomace extracts as a natural washing aid to mitigate enterohaemorrhagic E. coli in fresh produce

The manuscript entitled “Berry pomace extracts as a natural washing aid to mitigate enterohaemorrhagic E. coli in fresh produce” describes the antimicrobial effect of berry pomace extract against E. coli on spinach and lettuce and the effect on gene expression of genes related to adhesion and virulence. The manuscript is interesting but several parts need improvement/extensive revision (in particular materials and methods). Please follow the comments in the file. 

Comments on the Quality of English Language

Moderate changes are necessary. Please see the report.

Author Response

Please see the attachment. Both the reviewers' response is mentioned in the single document. Thank you.

Round 2

Reviewer 2 Report

Comments and Suggestions for Authors

Authors revised the original manuscript according to reviewer's comments. However, few parts should be improved prior the acceptance. Please see the comments below:

Please use square brackets for the references in the main text

L125-129 Ok. Which is the reduction in the bacterial load after UV treatment in your case? Please add this information in the manuscript

L136 Here and throughout the manuscript add the medium, the diluent used for the serial dilutions, and the incubation conditions.

L152-155 Rewrite. It is not correct in English

L244-246 Rewrite. It is not correct in English

L275-277 Please add that this is a limitation of this study and some attemps should be considered to improve the efficacy of the washing treatment with BPE (e.g., increase the concentration of BPE and prolong the time of exposure). Revise this part considering these aspects

L446-447 How the sterilization could be done? Add this information

L519-520 Delete "while economical in the sense that it is a by-product of juice processing industries"

L529-751 Revise the formatting of references according to the style of the Journal (see guide for authors of Foods).

Comments on the Quality of English Language

Moderate changes are required
